# Measurement and Correlation of the Solubility of β-Cyclodextrin in Different Solutions at Different Temperatures and Thermodynamic Study of the Dissolution Process

**Shanshan Jin [1] , Xuewei Cui [1], Yingping Qi [2], Yongfeng Shen [2] and Hua Li [1],***

[1]   School of Chemical and Energy Engineering, Zhengzhou University, Zhengzhou 450001, China; 15737953500@163.com (S.J.); 17625672284@163.com (X.C.)
[2]   Zhengzhou Museum, Zhengzhou 450002, China; 18717804222@163.com (Y.Q.); 15737953500@126.com (Y.S.)
*   Correspondence: lihua@zzu.edu.cn; Tel.: +86-136-5386-0337

**Abstract:** A new improved formulation was studied to improve the rehydration properties of freeze-dried dumplings. To provide basic data for industrial applications, the solubility capabilities of β-Cyclodextrin in sucrose, NaCl, and a mixed solution were measured at temperatures ranging from 303.15 to 353.15 K using a laser monitoring method. The experimental values indicated that the solubility of β-Cyclodextrin in solvents increased with increasing temperature. The simplified Apelblat model, Apelblat model, and $\lambda h$ model were employed to analyze the experimental results using correlation tests. The relative average deviation (RAD) values between the experimental and calculated values were less than 0.095, 0.075, and 0.103 for the simplified Apelblat equation, Apelblat equation, and $\lambda h$ equation, respectively. Apparent thermodynamic analysis of β-Cyclodextrin dissolution was also performed at the mean temperature using the model parameters of Apelblat equation. Furthermore, thermodynamic properties of the solution process, including the enthalpy, entropy, and Gibbs free energy, were also calculated and analyzed. This study provides the basic data for applications in industrial production, and is specifically of great significance for the improved formulation of freeze-dried dumplings.

**Keywords:** β-Cyclodextrin; solubility; simplified Apelblat equation; Apelblat equation; $\lambda h$ equation

## 1. Introduction

Dumplings are a traditional delicacy in China, and are also consumed all over the world. With the acceleration of a fast-paced life and higher living standards, instant dumplings have attracted more attention from consumers. Scientific studies on instant dumplings include high-temperature deep-fried [1–4], hot-air-dried dumplings [5,6] and freeze-dried [7–9]. The use of frying to produce convenient dumplings is detrimental to human health. The convenient dumplings produced by the drying method have poor color, excessive loss of nutrients, and a large change in flavor. The freeze-drying process is able to ensure higher levels of food vitamins, proteins, and other nutrients, particularly those that are not susceptible to loss of heat-sensitive components. This process also maximizes their original nutrients, and it can restrain the harmful effects of bacteria and enzymes—which prevents oxidation during dryings—better than traditional drying methods [10–14]. Nevertheless, the rehydration of freeze-dried instant dumplings is still far from optimal in peoples' fast-paced lives. Although the literature [8] improves the rehydration of freeze-dried dumplings, it will affect the taste of freeze-dried dumplings. Determining the proper rehydration ratio, a suitable

formulation containing 4% β-Cyclodextrin, 6% sucrose, and 8% NaCl was achieved (Supplementary Materials).

The gathering of solid-liquid equilibrium data and the establishment of a thermodynamic model are of great significance in research on rehydration mechanisms. However, there have been few reports on the solubility of β-Cyclodextrin [11]. Solubility data is indispensable in industrial designs and applications for freeze-dried products [15]. β-Cyclodextrin was a common one which was obtained by seven D-glucopyranose polymerization, shown in Figure 1 [11,16]. It is widely used in the food, pharmaceutical and cosmetic industries for its special cavity structure [17]. The solubility of β-Cyclodextrin in 6% sucrose solution, 8% NaCl solution, and a mixed solution at temperatures ranging from 303.15 to 353.15 K was measured using a laser monitoring method. The simplified Apelblat model, Apelblat equation, and λh equation were used to assess correlations among the experimental results. The accuracy of the experimental data was verified by different methods of variance analysis.

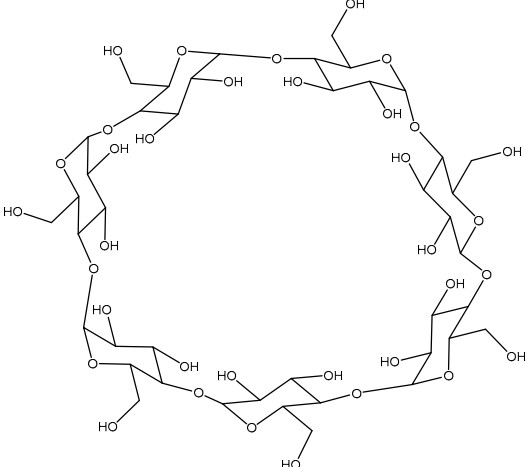

**Figure 1.** Chemical structure of β-Cyclodextrin.

## 2. Materials and Methods

### 2.1. Materials

β-Cyclodextrin was of analytical reagent grade and was obtained from the Shanghai Chemical Reagent Co., Ltd. (Shanghai, China). Sucrose and NaCl were of analytical reagent grade and were obtained from Tianjin Fengchuan Chemical Reagent Co., Ltd. (Tianjin, China). Deionized water was used in the experiments.

### 2.2. Solubility Measurement

Commonly used methods for solubility determination are the equilibrium method [18,19] and the dynamic method [20,21]. The solubility of β-Cyclodextrin in different solutions was investigated with the dynamic method under atmospheric pressure at temperatures ranging from 303.15 to 353.15 K. The determined procedure was carried out following the procedure reported in Reference [22]. The uncertainty of the temperature was ±0.02 K (calibrated using a standard thermometer).

The solubility of β-Cyclodextrin expressed in mole fraction was calculated as follows [18]:

$$x = \frac{m_1/M_1}{m_1/M_1 + m_2/M_2 + m_3/M_3} \tag{1}$$

where $m_1$ represents the mass of solute; $m_2$ and $m_3$ represent the mass of solute and solvent, respectively, in the sucrose solution; $M_1$, $M_2$, and $M_3$ are the molecular masses.

### *2.3. Test of Apparatus*

The solubility capabilities of NaCl and β-Cyclodextrin in water were measured and compared with the values reported in the literature to prove the feasibility and uncertainty in the measurements [23,24]. The experimental data agreed with the values reported, showing a mean relative deviation of 1.26% and 1.19%, respectively. Table 1 shows the measured values.

**Table 1.** Solubility of NaCl in water.

| T/K | x | x(lit) | 100RD |
|---|---|---|---|
| NaCl | | | |
| 293.15 | 0.1015 | 0.0999 | 1.58 |
| 313.15 | 0.1021 | 0.1011 | 0.98 |
| 333.15 | 0.1045 | 0.1033 | 1.15 |
| 373.15 | 0.1108 | 0.1094 | 1.26 |
| β-Cyclodextrin | | | |
| 293.15 | 0.0166 | 0.0164 | 1.20 |
| 313.15 | 0.0353 | 0.0349 | 1.13 |
| 333.15 | 0.0736 | 0.0729 | 0.95 |
| 353.15 | 0.1989 | 0.1966 | 1.50 |

## 3. Thermodynamic Models

1. The simplified Apelblat equation (ideal solution) is:

$$\ln x = A + B/T, \tag{2}$$

where A and B are the parameters of the model, x is the mole fraction solubility of β-Cyclodextrin, and T is the absolute temperature.

2. The Apelblat equation [25] is:

$$\ln x = A + \frac{B}{T} + C\ln T, \tag{3}$$

where x is the mole fraction solubility of β-Cyclodextrin, T is the absolute temperature, and A, B, and C are the model parameters.

3. The λh equation is:

$$\ln\left[1 + \frac{\lambda(1-x)}{x}\right] = \lambda h\left(\frac{1}{T} - \frac{1}{T_m}\right) \tag{4}$$

The variables λ and h are the model parameters derived from the experimental values, where x is the mole fraction solubility of β-Cyclodextrin at the system temperature T and $T_m$ is the normal melting temperature.

4. Deviation

The fitting accuracy with the equation was evaluated by the root mean square deviation (RMSD), relative deviation (RD), and relative average deviation (RAD):

$$RD = \frac{x_i - x_{ci}}{x_i} \tag{5}$$

$$RAD = \frac{1}{n}\sum_{i=1}^{n}\left|\frac{x_i - x_{ci}}{x_i}\right| \tag{6}$$

where *n* is the number of experimental points, $x_{ci}$ denotes the solubility determined from Equation (2), and $x_i$ denotes the experimental solubility value.

## 4. Results and Discussion

*4.1. The Molecule Thermodynamic Model of the Solubility of β-Cyclodextrin in Different Solutions at Different Temperatures*

The solubility capabilities of β-Cyclodextrin in sucrose, NaCl, and the mixed solution were investigated at different temperatures. Table 2 shows the RDs between the experimental values and computed values. The adjustable parameters of each model were fitted using the experimental solubility values with Origin software's non-linear regression method, the lsqnonlin function. The optimization procedure was established on $R^2$ (the square of the correlation coefficient). The optimized parameters and $R^2$ obtained by the Apelblat equation, simplified Apelblat equation, and $\lambda h$ equation are listed in Tables 3–5. Figure 1 shows the solubility curves through experimental points using the Apelblat equation.

**Table 2.** Mole fraction solubilities of β-Cyclodextrin in the sucrose, NaCl, and mixed solutions.

| T/K | $10^4 x_i$ | Simplified Apelblat | | Apelblat | | $\lambda h$ | |
|---|---|---|---|---|---|---|---|
| | | $10^4 x_{ci}$ | 100RD | $10^4 x_{ci}$ | 100RD | $10^4 x_{ci}$ | 100RD |
| | | | | 6% sucrose solution | | | |
| 303.35 | 4.06 | 3.12 | 23.15 | 3.43 | 15.51 | 5.14 | −26.60 |
| 313.15 | 6.44 | 5.49 | 14.75 | 5.72 | 11.23 | 7.11 | −10.40 |
| 322.95 | 9.36 | 9.31 | 0.53 | 9.42 | −0.63 | 9.96 | −6.41 |
| 333.05 | 15.39 | 15.56 | −1.10 | 15.45 | −0.39 | 14.51 | 5.71 |
| 343.35 | 24.03 | 25.44 | −1.41 | 25.22 | −4.96 | 22.50 | 6.37 |
| 353.05 | 40.17 | 39.39 | 1.94 | 39.53 | 1.58 | 37.75 | 6.02 |
| | | | | 8% NaCl solution | | | |
| 303.25 | 4.24 | 3.12 | 26.41 | 3.24 | 23.56 | 5.23 | −23.35 |
| 313.45 | 6.63 | 5.43 | 18.10 | 5.66 | 14.56 | 7.38 | −11.31 |
| 324.45 | 10.05 | 10.03 | 2.00 | 10.10 | −0.50 | 10.89 | −8.35 |
| 334.25 | 15.91 | 16.75 | −5.27 | 16.62 | −4.46 | 15.95 | −0.25 |
| 343.05 | 25.00 | 25.88 | −3.52 | 25.66 | −2.64 | 23.63 | 5.48 |
| 353.35 | 42.51 | 41.91 | 1.41 | 42.03 | 0.49 | 42.46 | 0.12 |
| | | | | mixture of 6% sucrose and 8% NaCl | | | |
| 303.35 | 4.22 | 3.33 | 21.09 | 3.82 | 9.43 | 5.16 | −22.27 |
| 313.45 | 6.54 | 5.77 | 11.77 | 6.21 | 4.97 | 7.21 | −10.24 |
| 323.05 | 9.71 | 9.42 | −5.17 | 9.72 | −0.08 | 10.06 | −3.60 |
| 333.65 | 14.9 | 15.67 | −5.17 | 15.72 | 5.49 | 14.98 | −0.54 |
| 343.15 | 23.09 | 24.08 | −4.29 | 23.88 | 3.46 | 22.56 | 2.30 |
| 352.95 | 37.18 | 36.60 | 1.56 | 36.36 | 2.18 | 38.19 | −2.72 |

**Table 3.** Parameters of the Apelblat equation for β-Cyclodextrin in the sucrose, NaCl, and mixed solutions.

| Solvent | A | B | C | $R^2$ | RAD |
|---|---|---|---|---|---|
| 6% sucrose | −107.98 | 381.57 | 17.28 | 0.995 | 0.056 |
| 8% NaCl | −95.09 | −428.53 | 15.48 | 0.995 | 0.075 |
| 6% sucrose and 8% NaCl | −98.63 | 275.95 | 15.72 | 0.996 | 0.043 |

**Table 4.** Parameters of the simplified Apelblat equation for β-Cyclodextrin in the sucrose, NaCl, and mixed solutions.

| Solvent | A | B | $R^2$ | RAD |
|---|---|---|---|---|
| 6% sucrose | 19.144 | −5461.89 | 0.994 | 0.071 |
| 8% NaCl | 19.784 | −5670.77 | 0.994 | 0.095 |
| 6% sucrose + 8% NaCl | 18.257 | −5173.19 | 0.994 | 0.082 |

**Table 5.** Parameters of the λh model for β-Cyclodextrin in the sucrose, NaCl, and mixed solutions.

| Solvent | λ | h | $R^2$ | RAD |
|---|---|---|---|---|
| 6% sucrose | 0.94 | 2409.01 | 0.996 | 0.103 |
| 8% NaCl | 0.94 | 2435.96 | 0.995 | 0.081 |
| 6% sucrose + 8% NaCl | 0.94045 | 2414.12 | 0.996 | 0.069 |

The RD between the computed and experimental values was less than 26%, indicating that the computed values had good agreement with the experimental results. From Tables 3–5, it can be seen that the RAD values were less than 0.095, 0.075, and 0.103 for the simplified Apelblat model, Apelblat model, and λ*h* model, respectively. This indicates that the three models agreed well with the experimental values.

The value of RAD was relatively small for the simplified Apelblat model, which shows that the data points were correlated with the simplified Apelblat model more closely than with the λh model. In addition, the Apelblat model with three adjustable parameters shows the better correlation with the experimental data compared to the simplified Apelblat model with two parameters.

As can be seen from Table 1 and Figure 2, the solubility capabilities of β-Cyclodextrin in sucrose, NaCl, and the mixed solution increased as the temperature rose. When the temperature was higher than 333.65 K, the solubility of β-Cyclodextrin in solvents was in order of 8% NaCl solution > 6% sucrose solution > the mixed solution. We concluded that the solvent composition and temperature influenced the solvent–solute interactions and that they affected the solubility.

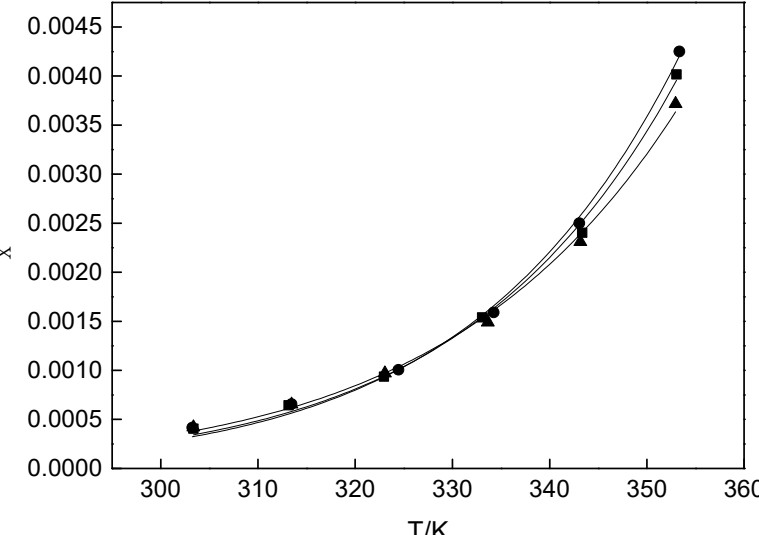

**Figure 2.** The solubility of β-Cyclodextrin in different solutions versus temperature with curve fitting using the Apelblat model. ■, 6% sucrose solution; ●, 8% NaCl solution; ▲, mixture solution of 6% sucrose and 8% NaCl.

### 4.2. Evaluation of the Model

To evaluate the goodness of fit between the experimental and predicted solubility data, the optimized model was employed for the prediction of the x data at 18 data points. Five general statistical parameters were employed in the constructed model to evaluate the predictive power of the model for x values. The experimental data $x_i$ from each prediction step sample was compared with the predicted $x_{ci}$. The residual sum of squares of prediction (PRESS) statistic seemed to be the most significant parameter. It accounted for a good estimate of the real predictive error of the models. The small value of the PRESS statistic suggested that the prediction of the model was better than chance and it was regarded as statistically significant. The PRESS statistic can be written as:

$$\text{PRESS} = \sum_{i=1}^{n}(x_{ci} - x_i)^2 \tag{7}$$

The predicted root mean square error (RMSEP) is a measurement of the average difference between predicted and experimental values at the prediction step [26]. RMSEP can be understood as the average prediction error, expressed in the same units as the original response values [26]. The RMSEP was calculated as:

$$\text{RMSEP}\left[\frac{1}{n}\sum_{i=1}^{n}(x_{ci} - x_i)^2\right]^{0.5} \tag{8}$$

The relative error of prediction (REP), the third statistical parameter, shows the predictive ability of each component, and was obtained with the following equation:

$$\text{REP}(\%) = \frac{100}{\overline{x}}\left[\frac{1}{n}\sum_{i=1}^{n}(x_{ci} - x_i)^2\right]^{0.5} \tag{9}$$

There are various ways to describe the predictive applicability of a regression model. The standard error of prediction (SEP) [26] was given by Equation (10), which is the most general expression:

$$\text{SEP} = \left[\frac{\sum_{i=1}^{n}(x_{ci} - x_i)^2}{n-1}\right]^{0.5}, \tag{10}$$

The $R^2$ is the square of the correlation coefficient; it was calculated as follows:

$$R^2 = \frac{\sum_{i=1}^{n}(x_{ci} - \overline{x})^2}{\sum_{i=1}^{n}(x_i - \overline{x})^2}, \tag{11}$$

where $x_{ci}$ represents the calculating data, $x_i$ is the experimental data, n is the total number of samples in the calculating set, and $\overline{x}$ is the mean of the experimental data. The statistical parameters were used to assess the models. The results are listed in Tables 6–8. By evaluating the data in Tables 6–8, the values of five statistical parameters, especially $R^2$ values and PRESS, were satisfactory. This indicated that the model agreed with the experimental data. For the Apelblat model, the PRESS value was relatively small. This showed that the data points correlated to the Apelblat model.

**Table 6.** Results of statistical parameters of the simplified Apelblat equation.

| Solvent | PRESS | RMSEP | REP (%) | SEP | $R^2$ |
|---|---|---|---|---|---|
| 6% sucrose | $4.4140 \times 10^{-8}$ | $8.57712 \times 10^{-5}$ | 5.174733 | $9.39577 \times 10^{-5}$ | 0.994 |
| 8% NaCl | $4.9188 \times 10^{-8}$ | $9.05428 \times 10^{-5}$ | 5.206601 | $9.91846 \times 10^{-5}$ | 0.994 |
| 6% sucrose + 8% NaCl | $3.3785 \times 10^{-8}$ | $7.50388 \times 10^{-5}$ | 4.707583 | $8.22009 \times 10^{-5}$ | 0.994 |

**Table 7.** Results of statistical parameters of the Apelblat equation.

| Solvent | PRESS | RMSEP | REP (%) | SEP | $R^2$ |
|---|---|---|---|---|---|
| 6% sucrose | $2.7482 \times 10^{-8}$ | $6.76781 \times 10^{-5}$ | 5.054669 | $7.41377 \times 10^{-5}$ | 0.995 |
| 8% NaCl | $3.1135 \times 10^{-8}$ | $7.20359 \times 10^{-5}$ | 4.142375 | $7.89113 \times 10^{-5}$ | 0.995 |
| 6% sucrose + 8% NaCl | $1.5655 \times 10^{-8}$ | $5.10899 \times 10^{-5}$ | 3.205138 | $5.59553 \times 10^{-5}$ | 0.996 |

**Table 8.** Results of statistical parameters of the $\lambda h$ equation.

| Solvent | PRESS | RMSEP | REP (%) | SEP | $R^2$ |
|---|---|---|---|---|---|
| 6% sucrose | $10.947 \times 10^{-8}$ | $1.3507 \times 10^{-4}$ | 8.149264 | $1.47966 \times 10^{-4}$ | 0.996 |
| 8% NaCl | $4.1292 \times 10^{-8}$ | $0.8285 \times 10^{-4}$ | 4.770432 | $0.90875 \times 10^{-4}$ | 0.995 |
| 6% sucrose+8% NaCl | $2.7624 \times 10^{-8}$ | $0.6785 \times 10^{-4}$ | 4.256761 | $0.74328 \times 10^{-4}$ | 0.996 |

*4.3. Thermodynamic Study of the Dissolution Process of β-Cyclodextrin in Different Solutions*

Thermodynamic functions are mainly dissolving enthalpy ($\Delta H$), entropy ($\Delta S$) and Gibbs free energy ($\Delta G$) related to the dissolution process, and can be obtained through a series of thermodynamic equation. The dissolution process of solid (S) in liquid (W) can be expressed as S + W = SW, and the relation between dissolution equilibrium and activity can be expressed as:

$$K_i = \frac{a_i}{a_s \times a_w} \tag{12}$$

where $a_i$ represents the activity of β-Cyclodextrin in a solution, and $a_s$ and $a_w$ represent activity of pure solids and liquids, respectively. In general, pure solid and liquid are considered as standard states, so $a_s$ and $a_w$ are considered as constants. The relationship between activity $a_i$ and mole fraction $x_i$ can be expressed as:

$$a_i = \gamma_i \times x_i \tag{13}$$

Therefore, Equation (12) can be expressed as:

$$K_i = \frac{\gamma_i \times x_i}{a_s \times a_w} \tag{14}$$

where $\gamma_i$ is the activity coefficient of β-Cyclodextrin. Under the assumption that $\gamma_i$ is a constant value, Equation (14) can be converted into:

$$\ln K_i = \ln x_i + J \tag{15}$$

where $J = \ln \gamma_i - \ln(a_s \times a_w)$, which is a temperature-independent constant.

According to the Gibbs equation and correction of Van 't Hoff equation, dissolving enthalpy ($\Delta H$) can be expressed as:

$$\Delta H = -R \times \frac{d\ln K_i}{dT^{-1}} \tag{16}$$

$$\Delta H = -R \times \frac{d\ln x_i}{dT^{-1}} \tag{17}$$

Substituting Equation (3) into Equation (17), the following equation was obtained [27]:

$$\Delta H = R \times T \times \left(C - \frac{B}{T}\right) \tag{18}$$

According to the relation between thermodynamics enthalpy, entropy and Gibbs free energy [28,29], we obtained:

$$\Delta S = R(A + C + C\ln T) \tag{19}$$

$$\Delta G = -RT\left(A + \frac{B}{T} + ClnT\right) \tag{20}$$

where A, B, and C represent the parameters gained by regression with the Apelblat model.

According to the modified model parameters by Equations (18) and (19), the $\Delta H$ and $\Delta S$ are listed in Table 9

**Table 9.** $\Delta H$ and $\Delta S$ of β-Cyclodextrin in different solutions.

| T (K) | $\Delta H$ (KJ·mol$^{-1}$) | $\Delta S$ (J·mol$^{-1}$·K$^{-1}$) | $\Delta G$ (KJ·mol$^{-1}$·K$^{-1}$) |
|---|---|---|---|
| β-Cyclodextrin in 6% sucrose solution | | | |
| 303.35 | 40.407 | 66.95 | 20.09 |
| 313.15 | 41.816 | 71.52 | 19.42 |
| 322.95 | 43.224 | 75.95 | 18.69 |
| 333.05 | 44.675 | 80.37 | 17.90 |
| 343.35 | 46.154 | 84.75 | 17.06 |
| 353.05 | 47.548 | 88.75 | 16.21 |
| β-Cyclodextrin in 8% NaCl solution | | | |
| 303.25 | 42.593 | 73.59 | 20.27 |
| 313.45 | 43.906 | 77.85 | 19.53 |
| 324.45 | 45.322 | 82.29 | 18.75 |
| 334.25 | 46.583 | 86.12 | 17.89 |
| 343.05 | 47.716 | 89.46 | 16.99 |
| 353.35 | 49.041 | 93.27 | 16.11 |
| β-Cyclodextrin in a mixture of 6% sucrose and 8% NaCl | | | |
| 303.35 | 37.357 | 57.56 | 24.47 |
| 313.45 | 38.677 | 61.88 | 23.88 |
| 323.05 | 39.932 | 66.39 | 23.26 |
| 333.65 | 41.317 | 70.28 | 22.57 |
| 343.15 | 42.559 | 73.67 | 21.83 |
| 352.95 | 43.840 | 77.54 | 21.10 |

From Table 9, dissolving enthalpy ($\Delta H$) and entropy ($\Delta S$) of β-Cyclodextrin in different solutions are positive, and increase with the rising of temperature. Gibbs free energy ($\Delta G$) represents the minimum energy that is required to dissolve β-Cyclodextrin under the experimental conditions. The values of $\Delta G$ are positive. Therefore, the solution process is always endothermic, which explains the increasing β-Cyclodextrin solubility with increasing temperature.

## 5. Conclusions

The solubility capabilities of β-Cyclodextrin in sucrose, NaCl, and the mixed solution were measured at temperatures ranging from 303.15 to 353.15 K. Computational analysis indicated that the computed values had good agreement with the experimental results. The results showed that the Apelblat equation was better than the simplified Apelblat equation and the λh equation. The thermodynamic properties including dissolution enthalpy, entropy, and Gibbs free energy were also obtained using the Apelblat model and the Van 't Hoff equation. The thermodynamic parameters values proved that the solution processes of β-Cyclodextrin in 6% sucrose, 8% NaCl and the mixture of 6% sucrose and 8% NaCl are endothermic. The correlated and experimentally derived solubility data of β-Cyclodextrin in this work can be used as a reference for research on industrial applications and to improve the formulation.

**Supplementary Materials:** The following are available online at http://www.mdpi.com/2227-9717/7/3/135/s1, Figure S1: The image of hot-air dried dumplings, Figure S2: The image of freeze-dried dumplings, Figure S3: Effect of the concentration of sucrose on rehydration ratio of freeze-dried dumplings, Figure S4: Effect of the concentration of β-Cyclodextrin on rehydration ratio of freeze-dried dumplings, Figure S5: Effect of the concentration of NaCl on rehydration ratio of freeze-dried dumplings.

**Author Contributions:** H.L. contributed to the conception of the study. X.C. contributed significantly to analysis and manuscript preparation. S.J. performed the data analyses and wrote the manuscript. Y.Q. and Y.S. helped perform the analysis with constructive discussions.

**Funding:** This research received no external funding.

**Conflicts of Interest:** We have no conflicts of interest to declare.

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
