# Peer review of "Measurement and Correlation of the Solubility of β-Cyclodextrin in Different Solutions at Different Temperatures and Thermodynamic Study of the Dissolution Process"

_processes, doi:10.3390/pr7030135_

Round 1
Reviewer 1 Report
Manuscript reports the solubility of the solubility of Β-Cyclodextrin in sucrose, NaCl, and a mixed solution was measured at temperatures ranging from 303.15–353.15 K using a laser monitoring method. The variation in the mole fraction solubility with temperature was mathematically correlated using the simplified Apelblat model, Apelblat model, and λh model. The relative average deviation (RAD) between the experimental and back-calculated values was less than 0.095, 0.075, and 0.103 for the simplified Apelblat equation, Apelblat equation, and λh equation, respectively. As part of the study the authors also calculated the apparent thermodynamic analysis of β-Cyclodextrin dissolution at the mean temperature using the model curve-fit parameters of Apelblat equation.
The apparent entropies of dissolution are miscalculated. I know that the authors likely obtained the equation that they are using (namely equation 20) from the published literature. Equation 20 is incorrect, however. The authors might wish to read the following three papers:
1: L. Liu, H. Li, D. Chen, X. Zhou, Q. Huang, H. Yang, Solubility of 1,1-diamino-2,2-dinitroethylene in different pure solvents and binary mixtures (dimethyl sulfoxide + water) and (N,N-dimethylformamide + water) at different temperatures, Fluid Phase Equilib. 460 (2018) 95–104.
2: L. Zhu, L.-Y.Wang, X.-C. Li, Z.-L. Sha, Y.-F.Wang, L.-B. Yang, Experimental determination and correlation of the solubility of 4-hydroxy-2,5-dimethyl-3(2H)-furanone (DMHF) in six different solvents, J. Chem. Thermodyn. 91 (2015) 369–377.
3: W. E. Acree Jr., F. Martínez, Comment on Measurement, Correlation, and Thermodynamic Properties for Solubilities of Bioactive Compound (-)-Epicatechin in Different Pure Solvents at 298.15 K to 338.15 K, J. Mol. Liq. 266 (2018) 441-442.
References 7 and 8 need to be translated to English.
The authors need to perform a more complete search of the published literature to find experimental solubility data to compare their measured values to. I am sure that the published literature contains several studies that have reported the solubility of beta-cyclodextrin in water.
Author Response
Point 1: The apparent entropies of dissolution are miscalculated. I know that the authors likely obtained the equation that they are using (namely equation 20) from the published literature. Equation 20 is incorrect, however. The authors might wish to read the following three papers:
1: L. Liu, H. Li, D. Chen, X. Zhou, Q. Huang, H. Yang, Solubility of 1,1-diamino-2,2-dinitroethylene in different pure solvents and binary mixtures (dimethyl sulfoxide + water) and (N,N-dimethylformamide + water) at different temperatures, Fluid Phase Equilib. 460 (2018) 95–104.
2: L. Zhu, L.-Y.Wang, X.-C. Li, Z.-L. Sha, Y.-F.Wang, L.-B. Yang, Experimental determination and correlation of the solubility of 4-hydroxy-2,5-dimethyl-3(2H)-furanone (DMHF) in six different solvents, J. Chem. Thermodyn. 91 (2015) 369–377.
3: W. E. Acree Jr., F. Martínez, Comment on Measurement, Correlation, and Thermodynamic Properties for Solubilities of Bioactive Compound (-)-Epicatechin in Different Pure Solvents at 298.15 K to 338.15 K, J. Mol. Liq. 266 (2018) 441-442.
Response 1: We have revised the Equation 20 and the apparent entropies of dissolution, in line 204, and Table 9. (in red)
Point 2: References 7 and 8 need to be translated to English.
Response 2: We have translated references 7 and 8 to English. (in red)
Point 3: The authors need to perform a more complete search of the published literature to find experimental solubility data to compare their measured values to. I am sure that the published literature contains several studies that have reported the solubility of beta-cyclodextrin in water.
Response 3: We have measured the solubility of β-Cyclodextrin in water and compared with the values reported in the literature. In Table 1. (in red)
Reviewer 2 Report
The authors report experimental measurements regarding the temperature dependent solubility of beta-cyclodextrin in different solutions. The experimental data are of interest for a broader community. I therefore recommend publication after the following points have been addressed:
(1) The section numbering is confusing. In line 108 section 4 starts, while later on line 145 the heading reads 3.3 and later on line 212 again section 4 is used.
(2) On line 123 it is mentioned that the Apelblat model shows the best correlation with the experimental data. However, this is not surprising given that this model as three adjustable parameters compared to two parameters for the other two models. Perhaps the authors should discuss this point.
(3) On line 128 the ranking of solubility described in the text is: 6% sucrose > 8% sucrose > NaCl solution, whereas Figure 2 suggests the ranking 8% sucrose > 6% sucrose > NaCl solution. The authors should clarify this discrepancy.
(4) On line 196 J is defined as temperature independent constant. Is that an assumption or why is J not temperature dependent? Please explain.
(5) On line 201 it should read: “Substitute (3) into (18)”
Author Response
Point 1: The section numbering is confusing. In line 108 section 4 starts, while later on line 145 the heading reads 3.3 and later on line 212 again section 4 is used.
Response 1: We have revised the section numbering, in line 145,181 and 214. (in red)
Point 2: On line 123 it is mentioned that the Apelblat model shows the best correlation with the experimental data. However, this is not surprising given that this model as three adjustable parameters compared to two parameters for the other two models. Perhaps the authors should discuss this point.
Response 2: The simplified Apelblat model as two adjustable parameters more closely than the λh model as two adjustable parameters. In addition, the Apelblat model as three adjustable parameters shows the better correlation with the experimental data compared to two parameters for the simplified Apelblat model(ideal solution) (in red)
Derivation process is shown in word (see attached word)
Point 3: On line 128 the ranking of solubility described in the text is: 6% sucrose > 8% sucrose > NaCl solution, whereas Figure 2 suggests the ranking 8% sucrose > 6% sucrose > NaCl solution. The authors should clarify this discrepancy.
Response 3: We have revised this mistake. See “line 128”.(in red)
Point 4: On line 196 J is defined as temperature independent constant. Is that an assumption or why is J not temperature dependent? Please explain.
Response 4: J=lnγi-ln(as*aw),as and aw represent activity of pure solids and liquids respectively. In general, pure solid and liquid are considered as standard states, so as and aw are considered as constants. See “line187-189,196”. (in red)
Point 5: On line 201 it should read: “Substitute (3) into (18)”
Response 5: We have revised this mistake. See “line 201”.(in red)
Round 2
Reviewer 1 Report
The authors have addressed my earlier comments.